# Inhibition of Insulin-like Growth Factor 1 Receptor/Insulin Receptor Signaling by Small-Molecule Inhibitor BMS-754807 Leads to Improved Survival in Experimental Esophageal Adenocarcinoma

**DOI:** 10.3390/cancers16183175

**Published:** 2024-09-17

**Authors:** Md Sazzad Hassan, Chloe Johnson, Saisantosh Ponna, Dimitri Scofield, Niranjan Awasthi, Urs von Holzen

**Affiliations:** 1Department of Surgery, Indiana University School of Medicine, South Bend, IN 46617, USA; nawasthi@iu.edu (N.A.);; 2Harper Cancer Research Institute, South Bend, IN 46617, USA; 3Department of Chemistry and Biochemistry, University of Notre Dame, Notre Dame, IN 46556, USA; 4Department of Biology, Indiana University, South Bend, IN 47405, USA; 5Goshen Center for Cancer Care, Goshen, IN 46526, USA; 6School of Medicine, University of Basel, 4056 Basel, Switzerland

**Keywords:** esophageal adenocarcinoma, BMS-754807, nanoparticle albumin-bound paclitaxel, apoptosis, in vitro wound healing, antitumor efficacy, survival benefit

## Abstract

**Simple Summary:**

Esophageal adenocarcinoma (EAC) is one of the deadliest malignancies in the United States (US) and the only major cancer in the US with an increasing incidence. Although recent advances have been made in surgery, radiation, and systemic medications, outcomes remain poor. Therefore, new beneficial treatment approaches are crucially desired. Obesity is associated with EAC and abnormalities in insulin-like growth factor-1 (IGF-1) and insulin signaling. However, the effect of blocking IGF-1 receptor/insulin receptor (IGF-1 R/IR) signaling is not well studied in EAC. Here, BMS-754807, a robust small-molecule inhibitor of IGF-1R/IR signaling, inhibited not only EAC cell growth but also EAC cell migration both as a monotherapy and in combination with the chemotherapy nanoparticle albumin-bound paclitaxel (nab-paclitaxel). In addition, BMS-754807 along with nab-paclitaxel significantly reduced EAC mice tumor growth, with enhanced mice survival. Interestingly, the combination of BMS-754807 and nab-paclitaxel showed an additive effect on tumor growth inhibition and survival benefit, indicating this combination as a novel option for EAC therapy.

**Abstract:**

The insulin-like growth factor-1 (IGF-1) and insulin axes are upregulated in obesity and obesity-associated esophageal adenocarcinoma (EAC). Nanoparticle albumin-bound paclitaxel (nab-paclitaxel) is a contemporary nanotechnology-based paclitaxel (PT) bound to human albumin, ensuring its solubility in water rather than a toxic solvent. Here, we examined the benefits of inhibiting insulin-like growth factor-1 receptor/insulin receptor (IGF-1/IR) signaling and the enhancement of nab-paclitaxel effects by inclusion of the small-molecule inhibitor BMS-754807 using both in vitro and in vivo models of EAC. Using multiple EAC cell lines, BMS-754807 and nab-paclitaxel were evaluated as mono and combination therapies for in vitro effects on cell proliferation, cell death, and cell movement. We then analyzed the in vivo anticancer potency with survival improvement with BMS-754807 and nab-paclitaxel mono and combination therapies. BMS-754807 monotherapy suppressed in vitro cell proliferation and wound healing while increasing apoptosis. BMS-754807, when combined with nab-paclitaxel, enhanced those effects on the inhibition of cell proliferation, increment in cell apoptosis, and inhibition of wound healing. BMS-754807 with nab-paclitaxel produced substantially greater antitumor effects by increasing in vivo apoptosis, leading to increased mice survival compared to those of BMS-754807 or nab-paclitaxel monotherapy. Our outcomes support the use of BMS-754807, alone and in combination with nab-paclitaxel, as an efficient and innovative treatment choice for EAC.

## 1. Introduction

Esophageal cancer (EC) has two main subtypes: esophageal squamous cell carcinoma (ESCC) and esophageal adenocarcinoma (EAC) [1]. Although ESCC still accounts for most cases of EC worldwide, EAC has become the main subtype in the United States (US) [2,3,4,5]. EAC incidence has been rapidly increasing in the West over the past few decades [2,3,4,5,6]. EAC primarily affects white males and is contributing to an increasing death rate among males in the US [7,8]. One-third to one-half of EAC patients treated with chemoradiation therapy or additional surgery survive only two years without recurrence of cancer [9,10,11]. Advanced-stage EAC, which has already spread from its primary site or has recurred after initial treatment, is usually very aggressive. These aggressive cancers often have bad outcomes, with overall five-year survival rates below 20 percent [12]. This is mainly due to high resistance to conventional chemotherapies and limited promising treatments [13]. Hence, novel treatment options are crucially needed to improve EAC patients’ clinical outcomes.

One of the risk factors for EAC is obesity [14]. Insulin/insulin-like growth factor-1 (IGF-1) signaling has been shown to be a critical arbitrator in obesity-related EAC [15,16,17,18]. Insulin and IGF-1 both have very important functions in relation to cell growth, differentiation, migration, and apoptosis [19,20,21]. Insulin receptor (IR) and insulin-like growth factor-1 (IGF-1R) are both critical in the progression of cancer, particularly in tumorigenesis and the development of cancer drug resistance due to cross-talk between IGF-1R and IR signaling pathways [21,22,23,24]. IGF signaling molecules primarily include the ligands IGF-1, IGF-2, and insulin and their receptors IGF-1 receptor (IGF-1R), insulin receptor (IR), and hybrid receptors [25]. The main carcinogenic responses of IGF engage IGF-1 and IGF-2 ligands binding to the IGF-1R, with the subsequent activation of downstream signals [26]. The activation of IR as well as IGF-1R/IR hybrid receptors by insulin may contribute to cancer growth, leading to resistance to IGF-1R inhibition [24]. As the IGF and insulin signaling axes have a crucial role in EAC cell proliferation and cancer progression, targeting IGF-IR/IR might therefore be an encouraging anticancer therapeutic path for EAC [24,27].

Paclitaxel (PT) in combination with carboplatin is an accepted treatment option for advanced EAC [28]. PT has a high response rate in EAC patients, and its combination with carboplatin is a moderately well-tolerated and safe regimen [29]. However, PT requires emulsification with solvents that can cause severe toxicities leading to severe negative effects in patients. Nanoparticle albumin-bound paclitaxel (nab-paclitaxel, NPT) is an albumin-bound water-soluble nanoparticle form of PT without any Cremophor and risk of capillary blockage [30]. Human serum albumin incorporated in nab-paclitaxel acts as a natural solubilizer and disperses rapidly after intravenous administration. Albumin has a long half-life and, by binding to the albumin-specific receptor, nab-paclitaxel can accumulate in the tumor more effectively than paclitaxel by an enhanced permeability and retention (EPR) effect [31]. Nab-paclitaxel (NPT) has demonstrated higher effectiveness than PT in EAC [32]. The absence of Cremophor in free nab-paclitaxel leads to a decrease in neutropenia and an improvement in peripheral neuropathy [30]. This means that nab-paclitaxel can be given to patients in greater doses with a briefer infusion time. Nab-paclitaxel utilizes albumin properties to reversibly bind paclitaxel, transport it across the endothelium, and concentrate it in areas of tumors [30]. Nab-paclitaxel has nearly double the response rate and shows increased time to disease progression as well as improved survival as a second-line therapy in comparison to paclitaxel [30].

BMS-754807 is a pyrrolotriazine (pyrrolo[2,1-f][1,2,4]triazine) that was first manufactured at Bristol-Mayers Squibb Pharmaceutical Research Institute [33]. BMS-754807 is a potent, reversible, small-molecule ATP-competitive tyrosine kinase inhibitor of the IGF-1R/IR family kinases that can be administered orally [34]. It specifically inhibits the catalytic function of IGF-1R [33]. It has a minor effect on Aurora A/B, Met, Ron, and TrkA/B and an insufficient effect on PKA, PKC, Flt3, Lck, MK2, and other protein kinases [35]. Previous studies have shown the anticancer actions of BMS-754807 in many cancer types both in vitro as well as in vivo [35,36,37,38,39]. This agent inhibits the growth of a broad range of cancer cell lines, including mesenchymal, epithelial, and hematological tumor cell lines [35,36,37]. It has been demonstrated by us and others that this compound induces apoptosis in some cancer cell lines associated with an increase in the cleavage of poly ADP–ribose polymerase (PARP) and caspase 3 expression [35,36,37]. This drug has been previously tested in several clinical studies for its safety and antitumor efficacy (NCT00569036, NCT01525823, NCT00908024, NCT00788333, and NCT00793897).

In this investigation, we evaluated the usefulness of inhibiting IGF/insulin signaling and the improvement of nab-paclitaxel effects by blocking IGF-1R/IR with the addition of small-molecule BMS-754807 in experimental EAC.

## 2. Materials and Methods

### 2.1. Mice, Cell Lines, and Reagents

Four- to six-week-old female athymic nude and non-obese diabetic/severe combined immunodeficient (NOD/SCID) mice purchased from Charles River (Wilmington, MA, USA) were utilized in this investigation. All mice were housed in a specific pathogen-free environment. The mouse experiments were performed in agreement with the regulations of the University of Notre Dame (UND) Institutional Animal Care and Use Committee (IACUC) and accepted by the UND IACUC under protocol number 18-09-4843.

Human esophageal cancer cell lines (Flo-1, ESO26, OE19, OE33, KYSE-270, SK-GT-2, ESO51, OE21, and OACM5.1C) [40] were purchased from Sigma Aldrich (St. Lois, MO, USA). Flo-1, ESO26, OE19, OE33, SK-GT-2, ESO51, and OACM5.1C are EAC cell lines [40], while OE21 and KYSE-270 are ESCC cell lines [40]. The cells were expanded by adopting standard cell culture techniques, and multiple aliquots were stored in liquid nitrogen, using cryotubes for the future use of early passages of cell lines in our experiments. We used RPMI-1640 medium with L-glutamine (Gibco, Grand Island, New York, NY, USA) for culturing our cell lines, except for Flo-1 [40]. Flo-1 was cultured in DMEM medium (Gibco) [40]. All cell lines were cultured and maintained in a medium containing 10% fetal bovine serum (FBS) at 37 °C in a humidified incubator with 5% CO_2_ and 95% air [40]. The cells were used within six months of re-expansion in culture. BMS-754807 was bought from Active Biochem (Maplewood, NJ, USA) [35]. Nab-paclitaxel was purchased from the Goshen Center for Cancer Care pharmacy (Goshen, IN, USA). The vehicle for both BMS-754807 and nab-paclitaxel was PBS. The cell proliferation testing agent WST-1 was bought from Roche Diagnostic Corporation (Indianapolis, IN, USA) [41].

### 2.2. Cell Viability Assay

The colorimetric WST-1 assay was used to assess the cell viability of the EAC cell lines as previously described [32]. Flo-1, OE19, and SK-GT-2 EAC cells were plated at 5000 cells per well in a 96-well plate in RPMI1640 with L-glutamine (Gibco, Grand Island, New York, NY, USA) and 10% FBS. [32]. The medium was changed after 36 h with 2% FBS in phenol red-free RPMI-1640 medium [32]. For detecting a dose-response effect of BMS-754807, EAC cells were treated with 50 nM to 10 μM of BMS-754807. For detecting a combination effect, EAC cells were treated with 5 μM of NPT and 1 μM of BMS-754807. These concentrations of NPT and BMS-57807 were chosen based on our previous publications [32,35,37,41]. After 72 h, 10 μL of WST-1 was added to each well [32]. The cells were then incubated for an additional two hours [32]. An absorbance microplate reader was used to measure the absorbance at 450 nm [32].

### 2.3. Western Blot Analysis

Western blot analyses were performed as previously described [32,41,42]. The phosphorylation status of IGF-1R/IR protein was first evaluated in a group of EAC cell lines. EAC cells were treated with 5 μM of NPT and 10 μM of BMS-75807 (BMS) alone or in combination for 16 h. These concentrations of NPT and BMS-57807 as well as the 16 h treatment period were chosen based on our previous publications [32,35,37,41]. After 16 h, the cells were lysed, and cell lysates were prepared by collecting cells from culture dishes in a cold lysis buffer (20 mM HEPES, 150 mM NaCl, 1 mM EDTA, 0.5% Na^+^ deoxycholate, 1% Nonidet P-40, and 1 mM DTT, pH 7.4) containing protease and phosphatase inhibitor cocktails (both from Sigma-Aldrich, St. Louis, MO, USA) [41]. SDS-PAGE was used to separate proteins in the supernatants. The proteins were transferred to nitrocellulose membranes (Bio-Rad, Hercules, CA, USA) and then incubated for 20 h at 4 °C with the following antibodies [41]: IGF-1R (Catalog #3027) diluted 1:1000, phospho- IGF-1R (Tyr1135/1136)/IR (Tyr1150/1151; Catalog #3024) diluted 1:1000, cleaved caspase-3 (Asp 175) (Catalog #9661) diluted 1:500, cleaved PARP (Aps 214) (Catalog #5625) diluted 1:1000, phospho-AKT (Ser473) (Catalog #4060) diluted 1:1000, AKT (Catalog #4685) (Cell Signaling Technology, Beverly, MA, USA) diluted 1:1000, and β-actin (Catalog # A1978) (MilliporeSigma, Rockville, MD, USA) diluted 1:10,000. Membranes were then incubated for 1 hour at room temperature with the species-specific HRP-conjugated secondary antibodies (Pierce Biotechnologies, Santa Cruz, CA, USA) diluted at 1:1000 [41]. Specific bands were identified using the enhanced chemiluminescence reagent (ECL, Perkin Elmer Life Sciences, Boston, MA, USA) according to the manufacturer’s instructions [41] and quantified by densitometry using ImageJ, Version 1.54 (National Institutes of Health, Bethesda, MD, USA).

### 2.4. Scratch Wound Healing Assessment

Flo-1 EAC cells were used to assess in vitro migration by scratch wound healing as described previously, with some modifications [43,44]. Half a million Flo-1 cells were plated into each well of 12-well plates. A straight continuous scratch was made in the middle of each well with a 200 μL pipette tip when the cells covered a higher than 90% area of each well to make a manufactured wound in the cell layer. The floating cells produced from the scratch were cautiously rinsed away three times with PBS. The cells were then treated with 5 µM of NPT and 10 µM of BMS-754807 alone and in combination in a reduced serum medium (2% FBS) for 96 h. The images were captured using a microscope (EVOS imaging system, Invitrogen, CA, USA) at 0, 24, 48, 72, and 96 h. The gap space in the cell layer was quantified using ImageJ, Version 1.54 software (National Institutes of Health, Bethesda, MD, USA). The wound healing rate was determined using the following standard equation [43,44]: wound healing rate = [(At = 0 h − At = 24 or 48 or 72 or 96 h)/At = 0 h] × 100, where At = 0 h and At = 24, 48, 72, or 96 h represent the mean gap area readings at 0, 24, 48, 72, and 96 h after scratching, respectively [43,44].

### 2.5. Animal Studies

Four- to six-week-old female athymic nude mice were injected subcutaneously with 5 million OE19 EAC cells [40]. As soon as the mice had quantifiable tumors (100 mm^3^), assessments of subcutaneous tumor size were started [40]. Two weeks after OE19 cell inoculation, all mice had quantifiable tumors (100 mm^3^) [40]. Mice were then injected intraperitoneally as previously stated after randomly grouping the mice into 4 groups, with 5 mice in each group with the vehicle (100 µL of PBS), BMS-754807 (25 mg/kg in 100 µL of PBS, 5 times a week for 2 weeks) [35,41], and NPT (10 mg/kg in 100 µL of PBS, 2 times a week for 2 weeks) [32] alone or in combination. These in vivo doses of NPT and BMS-57807 were chosen based on our previous publications [32,35,37,41]. The same doses of NPT or BMS-57807 as well as treatment duration were used for both mono and combination therapy. The tumor diameters were measured twice a week with a slide caliper for four consecutive weeks [40]. Tumor volume (TV) was determined as (W^2^XL)/2, where W is the width and L is the length of the tumor [40]. Relative tumor volume (RTV) was determined according to the following equation: RTV = TV_n_/TV_0_, where TV_n_ is the tumor volume on the day of evaluation and TV_0_ is the tumor volume on the first day of evaluation [32]. Net tumor growth was determined by subtracting the tumor volume at the beginning of treatment from that at the end [40]. All throughout the animal experiment, mouse weight was determined twice a week [40]. On the final day, the mice were sacrificed, and tumors were collected by careful dissection, weighed, and fixed in 4% formaldehyde for future immunohistochemistry (IHC).

Four- to six-week-old female NOD/SCID mice were used for survival experiments, as formerly stated [40]. Briefly, 10 million OE19 EAC cells were inoculated into the peritoneal cavity [40]. Fourteen days later, the mice were divided randomly into four groups of five mice in each group [40]. Each group of mice then received the vehicle, NPT (10 mg/kg, twice in 7 days), and BMS-754807 (25 mg/kg in 100 mL PBS, 5 times in 7 days) either alone or in combination via intraperitoneal injection for 14 days [40]. The survival of mice was checked from the first day of therapy until death [40].

### 2.6. Immunohistochemistry (IHC)

Standard IHC procedures were followed to stain the xenograft tumor tissues [32]. Xenograft tumor tissue blocks were fixed in 4% paraformaldehyde followed by embedding in paraffin [32]. Then, 5 μm tissue sections were cut followed by deparaffinization and then rehydration [32]. Antigen retrieval was augmented by hot temperatures in 10 mM sodium citrate, pH 6.0 [32]. Tissue sections were then treated with 3% (*v*/*v*) hydrogen peroxide for 30 min to counteract the tissue’s interior peroxidase action prior to blocking. Tissue sections were then blocked with 3% regular horse serum in PBS for 30 min [32]. Cleaved caspase-3 immunostaining was achieved using an anti–cleaved caspase-3 primary antibody (1:400 dilution; catalog 9661; Cell Signaling Technology) [32]. Similarly, Ki-67 immunostaining was achieved using a polyclonal anti–Ki-67 (1:200 dilution; ab15580; Abcam, Boston, MA, USA) antibody [32]. Anti-rabbit IgG and a peroxidase detection system (ImmPRESS^®^ HRP Horse Anti-Rabbit IgG PLUS Polymer Kit, Peroxidase, catalog # MP-7801, Vector Laboratories, Burlingame, CA, USA) were used for immune signal detection, where positive cells were stained brown [45]. Slides were counterstained with modified Mayer’s hematoxylin followed by dehydration through graded ethanol and xylenes and finally mounted onto coverslips with Cytoseal XYL mounting medium (VWR catalog number 48212-196, Chicago, IL, USA) [32]. Stained images were captured into 5 high-power fields/slides for the quantification of proliferative and apoptotic indexes, as described by us previously [32]. Cancer cell proliferation was determined within the tumor by immunostaining of the tissue microsections for Ki67 nuclear protein. Brown-stained nuclei for Ki67-positive cells and blue-stained nuclei for the total number of cells within the tumor tissue microsections were counted in five different microscopic images of high-power fields, and data were summarized as bar graphs (proliferative index). In each high-power field image, the proliferative index was measured by the number of brown-stained Ki67-positive tumor nuclei as a percentage of the total number of nuclei. Cancer cell apoptosis was determined within the tumor by immunostaining of tumor tissue microsections for cleaved caspase 3 cytoplasmic protein. Brown-stained cleaved caspase 3 and blue-stained tumor nuclei were counted in five different microscopic images of high-power fields, and data were summarized as bar graphs (apoptosis index). In each high-power field image, the apoptosis index was measured by the number of brown-stained cleaved caspase 3 positive cells as a percentage of the total number of nuclei.

### 2.7. Statistical Test

In vitro cell viability, RTV, net tumor growth, xenograft tumor weights, mice body weights, proliferative index, apoptotic index, and wound healing percentage results are shown as the mean ± standard deviation. A two-tailed Student’s *t*-test was used to interpret statistical significance for individual group comparisons. The comparison of RTV between treatment groups was concluded initially by determining the RTV number at day 14 by the mean RTV number of the analogous group at day 0, and later adopting the two-sample *t*-test, applied in the “t.test” R power (Warsaw, Poland) [32]. The survival time differences between treatment groups were analyzed by the log-rank test with GraphPad Prism 7.0 software [32] (GraphPad Software, San Diego, CA, USA). A *p*-value of less than 0.05 was recognized as statistically significant.

## 3. Results

### 3.1. Effects of BMS-754807 Alone and in Combination with Nab-Paclitaxel on EAC Cell Proliferation and Apoptosis

A Western blot study of cell lysates prepared from seven human EAC cell lines, Flo-1, ESO26, OE19, OE33, SK-GT-2, ESO51, and OACM5.1C, and two human ESSC cell lines, KYSE-270 and OE21, demonstrated that all these cell lines express IGF-1R and phospho-IGF-1R/IR proteins, with the strongest phospho-IGF-1R/IR expression in Flo-1, OE19, and SK-GT-2 (Figure 1). An evaluation of in vitro cell viability of the three EAC cell lines (Flo-1, OE19, and SK-GT-2) with the strongest expression of phospho-IGF-1R/IR showed that both nab-paclitaxel and BMS-754807 suppressed EAC cell viability (Figure 2). BMS-754807 dose-dependently inhibited Flo-1, OE19, and SK-GT-2 EAC cell proliferation (Figure 2A–C). The inhibition of cell proliferation at 1 µM of BMS-754807 was 28%, 37%, and 23% in Flo-1, OE19, and SK-GT-2 cells, respectively (Figure 2D–F). This effect of BMS-754807 alone on the inhibition of in vitro EAC cell proliferation is comparable to that of pancreatic cancer cell lines, as observed in our previous studies [35]. However, BMS-754807 alone did not exhibit a very strong in vitro antiproliferative effect and may not be a very good choice for treating this cancer as a monotherapy. The combination of 5 μM of nab-paclitaxel (NPT) with 1 μM of BMS-754807 significantly enhanced the inhibitory effect (Figure 2D–F). Western blot analysis demonstrated that BMS-754807 decreased phospho-IGF-1R/IR expression, indicating its effect as an IGF signaling inhibitor in EAC cells, leading to the inhibition of phospho-AKT (pAKT), a downstream signaling protein of IGF in Flo-1, OE19, and SK-GT-2 EAC cell lines (Figure 3A–C). Both NPT and BMS-754807 increased the expression of pro-apoptotic proteins cleaved PARP and cleaved caspase 3 (c-caspase 3), indicating their role in inducing apoptosis in EAC cells (Figure 3A–I). Furthermore, the combination of nab-paclitaxel and BMS-754807 enhanced these pro-apoptotic proteins’ expression in EAC cell lines (Figure 3A–I).

In this study, we did not correlate the in vitro antiproliferative and apoptotic effects of BMS-754807 with the high and low phospho-IGF-1R/IR or IGF-1R/IR expression in EAC cell lines, as different cell lines may have many genetic and epigenetic variations and such correlations are not optimal [41].

### 3.2. Effects of BMS-754807 Alone and in Combination with Nab-Paclitaxel on EAC Cell Scratch Wound Closure

Cancer cell metastasis occurs through a complex multistep process including cell migration away from the primary tumor. A scratch wound closure assay was used to explore the effect of nab-paclitaxel and BMS-754807 alone and in combination on Flo-1 EAC cell migration. Both nab-paclitaxel (NPT) and BMS-754807 (BMS) significantly impaired the migration of Flo-1 EAC cells, and their combination (NPT + BMS) significantly enhanced that effect (Figure 4A,B). Wound closure rates of Flo-1 cells treated for 24, 48, 72, and 96 h with 5 μM nab-paclitaxel were 20.32 ± 1.02%, 25.03 ± 1.44, 29.05 ± 2.10, and 35.33 ± 3.12, respectively; those of cells treated with 10 μM BMS-754807 were 18.83 ± 1.13, 22.58 ± 1.43, 24.18 ± 1.25, and 28.75 ± 1.37, respectively (Figure 4B). Combination treatment with nab-paclitaxel and BMS-754807 for 24, 48, 72, and 96 h significantly enhanced the inhibition of wound healing percentage from that of the single agent to 5.48 ± 1.25, 6.30 ± 2.15, 7.83 ± 1.71, and 11.28 ± 2.16, respectively (Figure 4B). These results suggest a pro-metastatic role of IGF-1 signaling in esophageal adenocarcinoma that can be inhibited by BMS-754807.

### 3.3. Effects of BMS-754807 Alone and in Combination with Nab-Paclitaxel on EAC Xenograft Tumor Size

Two weeks after the injection of 5 million OE19 EAC cells, control mice exhibited rapid OE19 cell-derived subcutaneous xenograft tumor size increase for the subsequent 2 weeks of the therapy time (Figure 5A). BMS-754807 monotherapy displayed a remarkable tumor regression compared to that of the control group during the two-week therapy period (Figure 5A). In addition, BMS-754807 added with nab-paclitaxel showed a compelling enhancement effect of tumor regression during the two-week therapy period (Figure 5A). BMS-754807 alone exhibited a compelling reduction in relative tumor volume (RTV) by 74.96% compared to the control, and the addition of nab-paclitaxel to BMS-754807 also exhibited a significant increase in tumor regression as the tumor volume reduced to 15.32% compared to the control (Figure 5B). The average net tumor growth after 2 weeks was 978.62 ± 92.79 mm^3^ for the control, as determined by deducting the tumor volume on the first day of therapy from that on the last day. After BMS-754807 single-agent therapy, net tumor growth significantly reduced to 329.97 ± 84.72 mm^3^; after nab-paclitaxel single-agent therapy, it significantly reduced to 227.79 ± 36.17 mm^3^; and after nab-paclitaxel plus BMS-754807 combined therapy, it was drastically reduced to 133.461 ± 33.31 mm^3^ (Figure 5C). A comparable decrease in tumor weight was also noticed by BMS-754807 and nab-paclitaxel single-agent therapy, and the combination therapy led to an even greater reduction in tumor weight (Figure 5D). However, mouse body weight did not show any significant difference under any of these treatment conditions (Figure 5E). At the completion of the treatment regimens, the mean tumor weight for the various treatment groups was 0.72 ± 0.10 g for the control, 0.44 ± 0.12 gm for NPT, 0.51 ± 0.13 gm for BMS-754807, and 0.24 ± 0.09 gm for NPT + BMS-754807 (Figure 5D).

### 3.4. Effects of BMS-754807 Alone and in Combination with Nab-Paclitaxel on Proliferation and Apoptosis within the Tumor In Vivo

Immunohistochemistry (IHC) analyses were performed on tumor microsections gathered from OE19 xenograft tumor blocks after the conclusion of 14 days of therapy to uncover the potential processes underlying the anticancer response of NPT and BMS-754807 combination. Immunostaining of the tissues with an antibody that binds to nuclear antigen ki67, a marker of active cell proliferation, unveiled a compelling diminished count of ki67, brown-stained tumor cells, in 100 counts of DAPI blue-stained total tumor cells (proliferative index) in both the BMS-754807 (BMS) and nab-paclitaxel (NPT) treatment groups compared to that of the vehicle-treated (control) group (Figure 6A,B). Interestingly, combining BMS-754807 with nab-paclitaxel exhibited a significant decrease in the proliferative index compared to that of the BMS-754807 or nab-paclitaxel monotherapy (Figure 6A,B). BMS-754807 monotherapy resulted in a 27% decrease in intratumoral proliferation when compared to that of the vehicle (control) group (*p* < 0.05). The combination of BMS-754807 with NPT resulted in a 61% decrease in proliferative activity when compared to that of the vehicle (control) group (*p* < 0.05). IHC assessment of apoptosis within the tumor was conducted by evaluating the expression of cleaved caspase 3 in OE19 xenograft tumor tissue microsections and displayed a significant increase in the count of cancer cells with cleaved caspase 3 brown staining out of 100 DAPI-stained total counts of cancer cells (apoptosis index) after in vivo treatment with either BMS-754807 or NPT compared to that of the vehicle (control) group (Figure 6C,D). Checking the apoptosis within the tumor tissue microsections revealed a 2.62-times increment in the apoptosis index in the BMS-754807 single-agent treatment group (*p* < 0.05) and a 2.27-times increment in the apoptosis index in the NPT single-agent treatment group (*p* < 0.05) compared to that of the vehicle (control) group (Figure 6C,D). The combination of BMS-754807 with NPT showed a 4.39-times increment in the apoptosis index (*p* < 0.05) compared to that of the vehicle (control) group (Figure 6C,D). Thus, combining BMS-754807 with NPT exhibited a significant increase in the apoptosis index compared to that of BMS-754807 or NPT monotherapy (Figure 6A,B).

### 3.5. Effects of BMS-754807 Alone and in Combination with Nab-Paclitaxe on EAC Animal Survival

We assessed the outcome of BMS-754807 (BMS) alone and in combination with nab-paclitaxel (NPT) on mice survival using a novel mouse xenograft survival model developed and characterized by us [40]. In an OE19 murine xenograft study with immunodeficient NOD/SCID mice with OE19 EAC tumors widely dispersed in the entire peritoneal cavity, the median survival of the mice was 47 days in the vehicle (control) group (Figure 7). With BMS-754807 (BMS) administration as a single agent, the median survival of mice significantly escalated to 57 days (121.27% enhancement compared to the control) (Figure 7). NPT monotherapy also resulted in a significant increase in median survival to 68 days (144.68% enhancement compared to the control) (Figure 7). BMS-754807 (BMS) plus NPT dual administration manifested a significant survival advantage not only over the vehicle (control) but equally over BMS-754807 (BMS) or NPT single-agent administration (*p* = 0.0034 in BMS + NPT vs. control; *p* = 0.0021 in BMS + NPT vs. BMS; *p* = 0.0339 in BMS + NPT vs. NPT). The dual administration of BMS-754807 (BMS) and NPT showed an enhanced median survival extended to 85 days (185.85% enhancement compared to the control) (Figure 7).

## 4. Discussion

Esophageal adenocarcinoma (EAC) is amongst the most destructive human malignancies and is the only dominant malignancy in the US with growing frequency [4,9,46]. The combination of paclitaxel with carboplatin is a well-tolerated first-line chemotherapy treatment option for advanced, unresectable, or metastatic EAC, providing some clinical benefits with survival advantage [47]. Unfortunately, most EAC patients are inoperable at the time of diagnosis. While at the beginning they show a good therapeutic response to systemic therapy, by the end, most EAC patients eventually die from the recurrence of the disease [48].

With the recognition of new biomarkers for EAC [49], targeted therapies are gaining interest [50,51], and new treatment preferences have been developed by targeting these proteins in EAC combination therapies [51]. With recent advances in dissecting the molecular mechanisms of EAC advancement, several targetable pathways like human epidermal growth factor receptor (HER1 and HER2) [52,53] and the vascular endothelial growth factor (VEGF) [54] have been identified. The HER2 receptor inhibitor transtuzumab was the first FDA-approved targeted therapy for EAC [55]. Trastuzumab was demonstrated to inhibit the growth of EAC with enhanced survival benefits when combined with standard chemotherapy in HER2-positive EAC [56].

Similarly, hyperinsulinemia and high IGF have been implicated in the progression of several malignancies including esophageal cancer [57,58,59]. The crucial role of IGF signaling mechanisms in malignancies is mediated by several vital components such as IGF-1 and 2 ligands, IGF-1R and 2R receptors, and IGFBPs 1 to 6 [60]. IGF-1R and IR activate an intracellular signal primarily through the phosphatidylinositol 3-kinase-AKT (PI3K-AKT) pathway in human cancers leading to mitogenesis, invasion, and apoptosis protection [61]. IGF-1R overexpression has been shown in other cancers, while the blockade of IGF-1R stimulation suppresses the expansion and spread of a broad range of malignancies [27,57]. However, for maximum therapeutic potential, suppression of both IGF-1R and insulin receptor (IR) activations is required [62]. Targeting IGF-1R individually without IR could lead to rescue mechanisms for insulin, leading to cancer advancement within unsuccessful clinical trials [63,64,65]. Very little is known about IGF-1R/IR activation and its blockade in human EAC. We found frequent overexpression and activation of IGF-1R/IR receptors in EAC cells.

BMS-754807 targets both IGF-1R as well as IR, and, in this study, this drug was evaluated in EAC cells. IGF-1R/IR phosphorylation was noticed in a wide range of esophageal cancer cell lines under normal growth conditions. Stronger IGF-1R/IR phosphorylation was noticed in Flo-1, OE19, and SK-GT-2 EAC cell lines than that of other EAC cell lines we tested. The IGF-1R receptor is also expressed in these three EAC cell lines. IGF-1R/IR phosphorylation under normal growth conditions may indicate some autocrine and/or paracrine effect of IGF on IGF-1R/IR in these EAC cells. BMS-754807 as a monotherapy adequately and dose-dependently suppressed the cell viability of OE19, Flo-1, and SK-GT-2 EAC cell lines in vitro. In an EAC nude mice xenograft model, BMS-754807 significantly inhibited local tumor growth, leading to the prolongation of mouse survival as a single agent. In addition, when BMS-754807 was combined with nab-paclitaxel, BMS-754807 significantly enhanced in vitro antiproliferative effects, leading to the enhancement of in vivo antitumor effects, with improved animal survival benefits. BMS-754807 given both as a single agent and a combined agent caused the diminished phosphorylation of IGF-1R/IR, leading to the reduction of succeeding signaling protein phospho-AKT in all three EAC cell lines. Previous reports on other cancers have indicated that BMS-754807 is a robust suppressor of the AKT pathway [66,67], which is consistent with our results. The antiproliferative and antitumor effects of BMS-754807 alone or in combination likely result from an increase in apoptosis [35,37]. We also observed enhanced cleavage of the apoptosis-associated caspase-3 and PARP molecules. Detailed mechanistic evaluations of apoptosis after BMS-754807 alone and in combination with NPT might help, but, here, we have shown increased apoptosis through caspase 3 and PARP-1 activation pathways. In this study, activated caspase 3 resulting from the cleavage of caspase 3 at Asp175 giving rise to 17/19-kDa fragments was detected. The nuclear protein PARP-1 is a substrate for activated caspase 3 in caspase-dependent apoptosis. PARP-1 is cleaved by activated caspase 3, resulting in the formation of 24-kDa and 89-kDa fragments. In the present study, we detected the 89-kDa fragment. The 89-kDa fragment of PARP-1 is translocated to the cytoplasm, inducing apoptosis by releasing the apoptosis-inducing factor (AIF) from mitochondria. Another innovative finding of this study is that as a single agent, BMS-754807 could suppress scratch closure in a wound healing assay, and combining BMS-754807 with nab-paclitaxel enhanced that effect. Other studies have found a crucial role of IGF-IR in the motility and invasion of tumor cells in other cancers [68,69]. It is possible that BMS-754807 can negatively impact EAC cell migration and, potentially, EAC metastasis. This is the first report of the detailed in vitro as well as in vivo effects of BMS-754807 as a monotherapy and dual therapy with the chemotherapeutic agent nab-paclitaxel on cell proliferation, apoptosis, cell signaling, cell migration, tumor regression, and animal survival in experimental EAC.

Paclitaxel and carboplatin chemotherapies have been used as standard anticancer drugs for the treatment of esophageal cancer, and the higher efficacy of NPT over paclitaxel or carboplatin was observed by us in a preclinical study [32]. In clinical trials, patients receiving NPT plus carboplatin had a significantly higher overall response rate (ORR) and disease control rate (DCR) than patients receiving the standard care of treatment with paclitaxel plus carboplatin [70]. In this study, we evaluated the efficacy of combining BMS-754807 with NPT in experimental esophageal adenocarcinoma models. Clinical trials in various cancer patients have failed, with monoclonal antibodies targeting only IGF-1R without IR (25,26,27) due to the advancement of tumor growth, most likely arising from the escape mechanism involving insulin [63,64]. Thus, our data support that the concurrent inhibition of IGF-1R and IR by BMS-754807 may give a better clinical outcome in EAC patients. The successful treatment of EAC patients depends on novel combination therapies of two or more therapeutic agents [71]. Combination therapies usually have significantly less toxicity in cancer patients as lower therapeutic doses of each individual drug can be used [72]. In addition, combination therapies can augment the anticancer effects of single-drug therapy [72]. Thus, these data support that the combination of BMS-754807 with NPT may produce desired anticancer effects with less toxicity in EAC patients.

IGF-1R and IR signaling is important in the regulation of blood glucose [73]. Both IGF and insulin can bind to the insulin receptor to stimulate the peripheral utilization of glucose and decrease the hepatic output of glucose [74]. As BMS-754807 is a strong suppressor of the IGF-1R and IR group, a rise of blood glucose, insulin, and its fragment C-peptide was noticed after BMS-754807 administration in a clinical trial [75]. However, in that clinical trial, daily dosing of BMS-754807 was found to be safe, leading to a continuation of the study, and its effects on plasma glucose were manageable [75]. The most frequent drug-associated toxicities of BMS-754807 were hyperglycemia, nausea, hypoglycemia, anorexia, and diarrhea [75,76]. Hyperglycemia was reversible and treated by giving metformin orally. Hypoglycemia was treated by giving glucose orally [76]. The most common toxicities associated with nab-paclitaxel were neuropathy, neutropenia, anemia, and thrombocytopenia [70]. Toxicities associated with these drugs can be improved by adjusting doses, drug interruption, and close monitoring. In a past experimental study, the effect of BMS-754807 was assessed on blood glucose in mice by performing an oral glucose tolerance experiment [36]. In that study, only a small increment in blood glucose or insulin measurements was observed, with no animal mortality [36]. The doses of nab-paclitaxel and BMS-754807 in our animal studies were based on our previous publications and we adopted an adjusted dosing regimen in the present study to minimize toxicity. In our studies, we also observed no noticeable in vivo noxiousness of the drug in our mice during the whole period of experiments.

## 5. Conclusions

IGF-1R and insulin receptor (IR) signaling systems are frequently coactivated in EAC. This research demonstrates for the first time the mechanism-specific efficacy of dual IGF-1R/IR signaling inhibition with BMS-754807 as a mono and combined therapy with the chemotherapy drug nab-paclitaxel in experimental EAC. BMS-754807 as a monotherapy produced greater in vitro antiproliferative, pro-apoptotic, and anti-motility effects and in vivo antitumor efficacy by inducing enhanced in vivo apoptosis with survival benefit compared to the control. The addition of nab-paclitaxel with BMS-754807 further augmented these effects, making them significantly higher than those noticed after single-agent therapy. These findings support the importance of BMS-754807 in blocking dual pathways of IGF and insulin involving IGF-1R/IR alone and together with nab-paclitaxel as an efficient novel choice in the treatment of EAC.

## Figures and Tables

**Figure 1 cancers-16-03175-f001:**
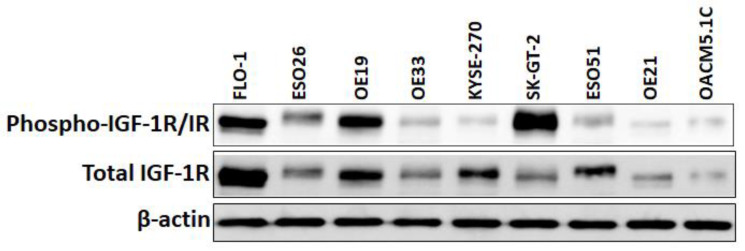
Phospho-IGF-1R/IR and total IGF-1R expression in human EC cell lines. Western blot experiment demonstrating phospho-IGF-1R/IR (95 kDa) and total IGF-1R (95 kda) expression in seven human EAC cell lines (FLO-1, ESO26, OE19, OE33, SK-GT-2, ESO51, and OACM5.1C) and two human ESCC cell lines (KYSE-270 and OE21). β-actin (42 kDa) served as a loading control. All these EC cell lines had IGF-1R and phospho-IGF-1R/IR expression, with the strongest phospho-IGF-1R/IR expression in FLO-1, OE19, and SK-GT-2. The uncropped blots are shown in the Appendix A.

**Figure 2 cancers-16-03175-f002:**
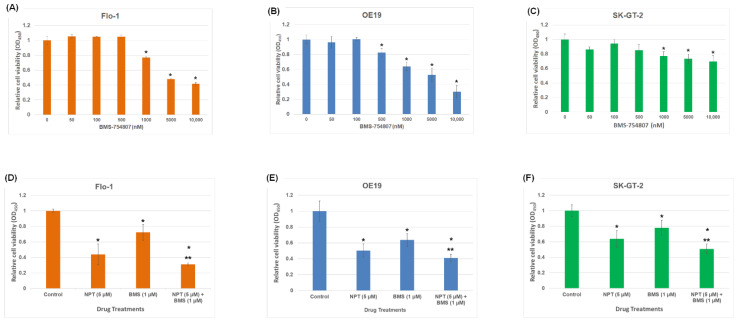
BMS-754807 only or in combination with a chemotherapy drug NPT induced cell proliferation inhibition in phospho-IGF-1R/IR-overexpressing EAC cells. (**A**–**C**) Dose-dependent effect of BMS-754807 on Flo-1, OE19, and SK-GT-2 cell proliferation inhibition. (**D**–**F**) BMS-754807 (BMS) showed enhancement of Flo-1, OE19, and SK-GT-2 cell proliferation inhibition when combined with the chemotherapy drug nab-paclitaxel (NPT). Flo-1, OE19, and SK-GT-2 EAC cells were seeded on 96-well plates and treated with (**A**–**C**) 0 nM to 10,000 nM concentrations of BMS-754807 alone and (**C**–**E**) a combination of 1 μM of BMS-754807 (BMS) and 5 µM of NPT simultaneously (NPT + BMS). After 72 h, the number of viable cells was analyzed by adding an equal amount of WST-1 reagent in each well, according to the manufacturer’s instructions. Results are the mean ± SDE of six values. * BMS-754807 additions significantly different from the control (0 nM). ** BMS + NPT additions significantly different from the single-agent addition (NPT or BMS). Results are representative of 3 separate experiments with similar results.

**Figure 3 cancers-16-03175-f003:**
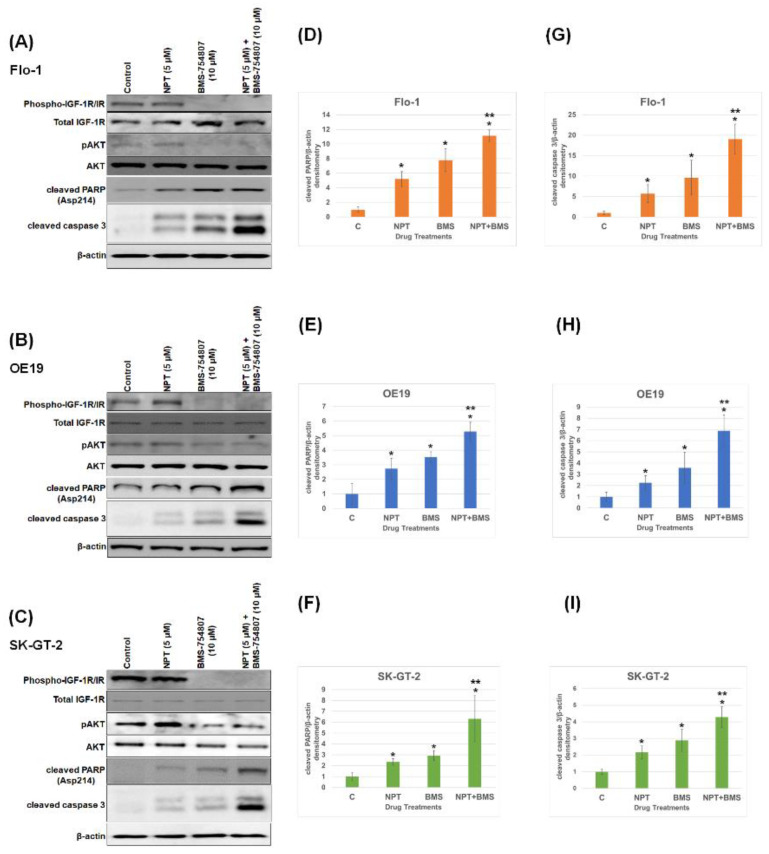
BMS-754807 only or in combination with the chemotherapy drug NPT induced the expression of apoptosis-related proteins through the inhibition of the IGF signaling pathway in phospho-IGF-1R/IR-overexpressing EAC cells. A total of 60 to 70% confluent human EAC cells (**A**) Flo-1, (**B**) OE19, and (**C**) SK-GT-2 in culture were given 5 μM of nab-paclitaxel (NPT) and 10 μM of BMS-754807 (BMS), either as a single agent or in combination, for a continuous 16 h. Total cell lysates were subjected to Western blotting with antibodies to phospho-IGF-1R/IR (95 kDa), total IGF-1R (95 kDa), phospho AKT (pAKT) (60 kDa), AKT (60 kDa), cleaved PARP (Asp214) (89 kDa), cleaved caspase 3 (Asp 175) (19, 17 kDa), and β-actin (42 kDa). The uncropped blots are shown in the Appendix A. The intensities of cleaved PARP (**D**–**F**) and cleaved caspase 3 (**G**–**I**) bands were quantified by densitometry and are represented in the bar graph, after normalizing values with β-actin. * NPT or BMS or BMS + NPT additions significantly different from the control (C). ** NPT + BMS additions significantly different from the single-agent addition (NPT or BMS) (N = 3).

**Figure 4 cancers-16-03175-f004:**
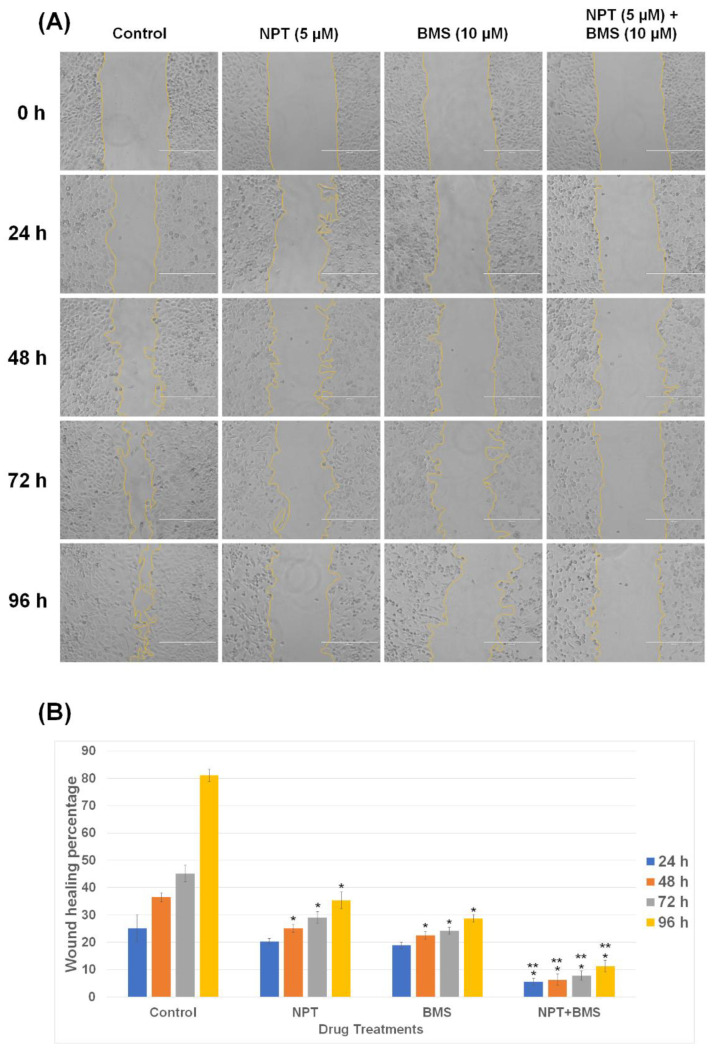
BMS-754807 alone or in combination with the chemotherapy drug NPT suppressed the migration of phospho-IGF-1R/IR-overexpressing Flo-1 EAC cells. (**A**) Near-confluent Flo-1 cells in culture were scratched with a pipette tip to make a linear gap and then 5 μM of nab-paclitaxel (NPT) and 10 μM of BMS-754807 (BMS) were added either as a single agent or in combination for a continuous 24, 48, 72, and 96 h (24 h, 48 h, 72 h, and 96 h). The microscopic photographs were taken at 0, 24, 48, 72, and 96 h, respectively (X10) (scale bar = 400 µm). Yellow lines indicate the borders of the scratched gap. (**B**) The gap area of Flo-1 cells was determined by Image J software to analyze the gap closure rate. All numbers were noted as the mean ± standard deviation (N = 5). * NPT or BMS-754807 (BMS) treatment significantly differed from the control (0 nM). ** NPT + BMS treatment significantly different from the single-agent therapy (NPT or BMS).

**Figure 5 cancers-16-03175-f005:**
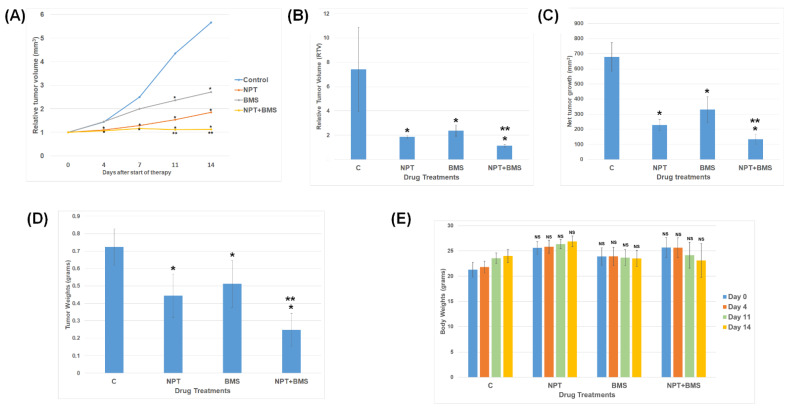
Phospho-IGF-1R/IR-overexpressing OE19 EAC xenograft tumor size reduction by BMS-754807 (BMS) only and in combination with the chemotherapy drug nab-paclitaxel (NPT). High phospho-IGF-1R/IR-expressing OE19 cells (5 million per site) were inoculated under the skin of nude mice. When measurable tumor size was reached (usually 2 weeks after cell inoculation), mice received the vehicle (**C**) and BMS and NPT alone or in combination. (**A**,**B**) Relative tumor volume (RTV) was determined by dividing the tumor volume at any time point by the tumor volume at the start of therapy. (**A**) RTV variation was observed over a duration of 14 days from the start of the therapy. (**B**) RTV variation was observed at the end of the therapy under different therapy regimens. (**C**) Variation in net growth in tumor volume was observed at the end of the therapy under different therapy regimens. It was determined by deducting the tumor volume determined on the 1st treatment day from that on the last day. (**D**) Average tumor weight variation was observed at the end of the therapy under different therapy regimens. (**E**) Variation in mouse body weight was observed during the 14-day therapy time. Results are average values ± standard deviation from 5 mice in each group. * Significantly different from control (**C**). ** Significantly different from single-drug therapy. NS = not significant.

**Figure 6 cancers-16-03175-f006:**
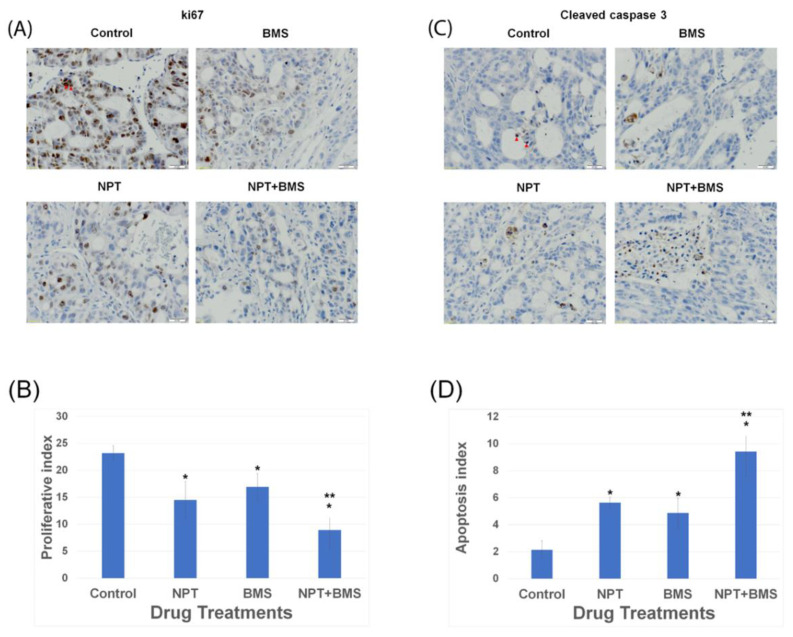
Anti-proliferative and pro-apoptotic in vivo efficacy of BMS-754807 monotherapy and in combination with chemotherapy. Xenografts developed in immunodeficient nude mice with high phospho-IGF-1R/IR-expressing OE19 EAC cells received either BMS-754807 (BMS) or nab-paclitaxel (NPT) monotherapy or BMS plus NPT combination therapy for 14 consecutive days. At the conclusion of therapy, tumors were collected and processed for immunohistochemistry (IHC). (**A**) Cancer cell proliferation was determined within the tumor by immunostaining of the tissue microsections for Ki67 nuclear protein. Representative microscopic high-power field images of Ki67 brown staining are shown. (scale bar = 20 µm). (**B**) Brown-stained nuclei for Ki67-positive cells and blue-stained nuclei for the total number of cells within the tumor tissue microsections were counted in five different microscopic images of high-power fields, and data are summarized as a bar graph (proliferative index). In each high-power field image, the proliferative index was measured by the number of brown-stained Ki67-positive tumor nuclei as a percentage of the total number of nuclei. (**C**) Cancer cell apoptosis was determined within the tumor by immunostaining of tumor tissue microsections for cleaved caspase 3 cytoplasmic protein. Representative microscopic high-power field images of cleaved caspase 3 brown staining are shown. (scale bar = 20 µm). (**D**) Brown-stained cleaved caspase 3 and blue-stained tumor nuclei were counted in five different microscopic images of high-power fields, and data are summarized as a bar graph (apoptosis index). In each high-power field image, the apoptosis index was measured by the number of brown-stained cleaved caspase 3-positive cells as a percentage of the total number of nuclei. Results are shown as the mean ± standard deviation. * Significantly different from vehicle (control). ** Significantly different from single-agent therapy.

**Figure 7 cancers-16-03175-f007:**
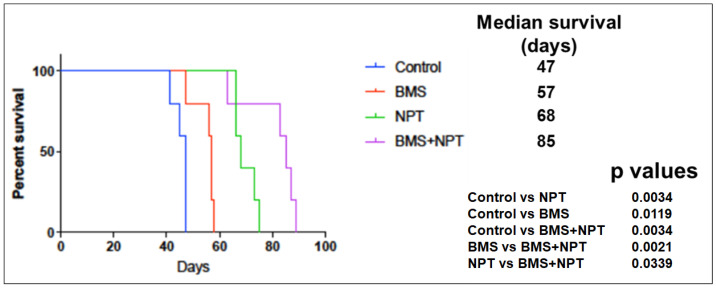
Mice survival enhancement by BMS-754807 (BMS) only and in combination with chemotherapy. Kaplan–Meier survival curve for mice harboring phospho-IGF-1R/IR-overexpressing OE19 EAC intraperitoneal xenograft treated for 14 days with BMS-754807 (BMS) only and in combination with the chemotherapy drug nab-paclitaxel (NPT). The curve shows mice survival in days from the start of drug administration. Statistical grouping variation in the survival period was conducted by log-rank measurement (GraphPad Prism 7.0).

## Data Availability

The data that support the findings of this study are available from the corresponding author upon reasonable request.

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
