# Peer review of "Inhibition of Insulin-like Growth Factor 1 Receptor/Insulin Receptor Signaling by Small-Molecule Inhibitor BMS-754807 Leads to Improved Survival in Experimental Esophageal Adenocarcinoma"

_cancers, 2024, doi:10.3390/cancers16183175_

Round 1
Reviewer 1 Report (Previous Reviewer 1)
Comments and Suggestions for Authors
The revision addressed my concerns to some extent, but I am still concerned about the lack of mechanistic distinction between IR and IGF-1R in the observed effects of BMS-754807. One can argue that for the purpose of treatment, the mechanistic distinction matters little if the combination works. However, the benefit of BMS-754807 seems fairly minor.
Author Response
Manuscript ID:
cancers-3179881
We thank the reviewers for reading the manuscript and providing constructive suggestions on how to improve it.
Reviewer 1:
Comments and Suggestions for Authors
The revision addressed my concerns to some extent, but I am still concerned about the lack of mechanistic distinction between IR and IGF-1R in the observed effects of BMS-754807. One can argue that for the purpose of treatment, the mechanistic distinction matters little if the combination works. However, the benefit of BMS-754807 seems fairly minor.
Responses:
The high sequence homology and cross-talk between the insulin receptor (IR) and insulin like growth factor-1R (IGF-1R) makes it challenging to dissect the mechanistic distinction between two signaling pathways, particularly when using small-molecule tyrosine kinase inhibitors (TKIs) like BMS-754807. As the reviewer pointed out, from a therapeutic standpoint, the precise mechanistic distinction may be less critical as long as there is demonstrable anti-tumor activity. This approach aligns with other studies indicating that if we block only one of them then the compensatory mechanism makes it less effective (references 63-64)). For example, monoclonal antibodies that specifically target either IR or IGF-1R tend to be less effective (references 24-27). For maximum therapeutic potential, suppression of both IGF-1R and IR activations are required. As we discussed in our manuscript targeting IGF-1R individually without IR could lead to rescue mechanisms for insulin, leading to cancer advancement within unsuccessful clinical trials. Regarding the benefit of BMS-754807, our in-vitro effects might be less pronounced as the drugs effect may reach at maximum achievable effect. But the in-vivo benefit of BMS-754807 was more pronounced.
Reviewer 2 Report (Previous Reviewer 3)
Comments and Suggestions for Authors
1. The description of nab-paclitaxel as a “nanotechnology-based paclitaxel bound to human albumin” lacks detail on how this formulation improves efficacy or reduces toxicity compared to conventional paclitaxel.
2. The MS does not address any potential side effects or toxicity associated with BMS-754807 or nab-paclitaxel, which is important for evaluating the overall safety of the treatment.
3. The MS mentions improved survival in mice, it lacks specific data or statistical significance related to survival outcomes.
4. The MS states that BMS-754807 and nab-paclitaxel increased apoptosis but does not describe the mechanisms or pathways involved in this process.
5. The article does not compare the efficacy of BMS-754807 and nab-paclitaxel to existing treatments for EAC.
Comments on the Quality of English Language
Minor editing of English language required.
Author Response
Manuscript ID:
cancers-3179881
We thank the reviewers for reading the manuscript and providing constructive suggestions on how to improve it.
Reviewer 2:
Comments and Suggestions for Authors
- The description of nab-paclitaxel as a “nanotechnology-based paclitaxel bound to human albumin” lacks detail on how this formulation improves efficacy or reduces toxicity compared to conventional paclitaxel.
Thank you for this important point. We have addressed this in our manuscript during our first revision. As suggested by the reviewer, we have elaborated it further in this revision. By adding following sentences in the manuscript.
(Line 84-90) Nanoparticle albumin-bound paclitaxel (nab-paclitaxel, NPT) is an albumin-bound water-soluble nanoparticle form of PT without any Cremophor and risk of capillary blockage [30]. Human serum albumin incorporated in nab-paclitaxel acts as natural solubilizer and dispersed rapidly after intravenous administration. Albumin has long half-life and by binding to albumin-specific receptor nab-paclitaxel can accumulate in the tumor more effectively than paclitaxel by enhanced permeability and retention (EPR) effect [31].
- The MS does not address any potential side effects or toxicity associated with BMS-754807 or nab-paclitaxel, which is important for evaluating the overall safety of the treatment.
Thank you for your valuable comment. The suggestion has been addressed. Following sentences have been added in the manuscript.
(Line 602-608) The most frequent drug associated toxicities of BMS-754807 were hyperglycemia, nausea, hypoglycemia, anorexia, and diarrhea [75, 76]. Hyperglycemia was reversible and treated by giving metformin orally. Hypoglycemia was treated by giving glucose orally [76]. The most common toxicities associated with nab-paclitaxel were neuropathy, neutropenia, anemia, and thrombocytopenia [70]. Toxicities associated with these drugs can be improved by adjusting doses, drug interruption and close monitoring.
(Line 611-613) The doses of nab-paclitaxel and BMS-754807 in our animal studies were based on our previous publications and we adopted an adjusted dosing regimen in the present study to minimize toxicity.
- The MS mentions improved survival in mice, it lacks specific data or statistical significance related to survival outcomes.
The survival study demonstrated a significant improvement in animal survival with BMS and NPT treatments. Importantly, combination therapy with BMS and NPT exhibited a significantly enhanced response. Specific data and statistical significance related to mice survival outcomes have been shown in new Figure 7 (Line 502). The percentage increase in survival in the BMS-754807 (BMS), NPT and BMS+NPT groups have been shown in the manuscript (Line 493, Line 495, Line 500) and the p-values are shown in new Figure 7 (Line 502). We have now included the percentage increase in survival and p values for the comparisons between the control group and the BMS and NPT groups.
- The MS states that BMS-754807 and nab-paclitaxel increased apoptosis but does not describe the mechanisms or pathways involved in this process.
Thank you for your comments. We agree with the reviewer that detailed mechanistic evaluation might help but, in this manuscript, we have described increased apoptosis through caspase 3 and PARP-1 activation pathways. The suggestion has been addressed. Following sentences have been added in the manuscript.
(Line 561-569) Detailed mechanistic evaluation in the apoptosis after BMS-754807 alone and in combination with NPT might help but here we have shown increased apoptosis through caspase 3 and PARP-1 activation pathways. In this study, activated caspase 3 resulting from cleavage of caspase 3 at Asp175 giving rise to 17/19-kDa fragments were detected. Nuclear protein PARP-1 is a substrate for activated caspase 3 in caspase dependent apoptosis. PARP-1 is cleaved by activated caspase 3 resulting in the formation of 24-kDa and 89-kDa fragments. In the present study, we detected 89-kDa fragment. 89-kDa fragment of PARP-1 is translocated to cytoplasm inducing apoptosis by releasing apoptosis inducing factor (AIF) from mitochondria.
- The article does not compare the efficacy of BMS-754807 and nab-paclitaxel to existing treatments for EAC.
Thank you for your comments. The suggestion has been addressed. Following sentences have been added in the manuscript.
(Line 579-585) Paclitaxel and carboplatin chemotherapies have been used as standard anticancer drugs for the treatment of esophageal cancer and higher efficacy of NPT over paclitaxel or carboplatin was observed by us in a preclinical study [32]. In clinical trials, patients receiving NPT plus carboplatin had significantly higher overall response rate (ORR) and disease control rate (DCR) than patients receiving standard care of treatment with paclitaxel plus carboplatin [70]. In this study, we evaluated the efficacy of combining BMS-754807 with NPT in experimental esophageal adenocarcinoma models.
Comments on the Quality of English Language
Minor editing of English language required.
As suggested by the reviewer, we have thoroughly reviewed the manuscript and made edits to the English language where necessary.
Round 2
Reviewer 1 Report (Previous Reviewer 1)
Comments and Suggestions for Authors
None.
Reviewer 2 Report (Previous Reviewer 3)
Comments and Suggestions for Authors
Recommended for publication.
Comments on the Quality of English Language
Minor editing of English language required.
This manuscript is a resubmission of an earlier submission. The following is a list of the peer review reports and author responses from that submission.
Round 1
Reviewer 1 Report
Comments and Suggestions for Authors
The paper by Hassan et al. examined the effects of BMS-754807 alone or in combination with chemotherapy on cell proliferation in vitro and tumor growth in animal models of esophageal adenocarcinoma. BMS-754807 is an inhibitor of IGF-1R and IR. The study demonstrated an inhibitory effect of this drug and in combination with nab-paclitaxel.
I have several concerns about this study that need to be addressed.
1. The title ignores the effect of BMS-754807 on insulin receptor, and just states the inhibition of insulin-like growth factor signaling, even though the text acknowledges the effect of the drug on insulin signaling in the study. The title is misleading. The text also sometimes focuses on IGF signaling to explain some effects without demonstrating it is IGF-1 rather than insulin. For example, Line 20-21 says “Here, BMS-754807 a robust small-molecule inhibitor of IGF signaling inhibited not only EAC cell growth”. Lines 277-278 states ”The above data suggested a pro-metastatic role of IGF-1 signaling in esophageal adenocarcinoma that can be inhibited by BMS-754807”. These statements are not accurate as BMS-754807 inhibition does not distinguish IGF-1 from insulin.
2. As an inhibitor of both IGF-1R and IR, BMS-754807 is expected to inhibit cell proliferation at least by blocking insulin signaling and energy metabolism. The inhibitory effects have been well documented in many cancer cell lines. This study does not demonstrate whether BMS-754807 also inhibits cell proliferation by blocking IGF-1R function. This distinction is mechanistically important. This was not demonstrated, even though the title claims the IGF-1R inhibition.
3. The study focused on three cell lines, Flo-1, OE19, and SK-GT-2. The reason for choosing these three cell lines were not stated, even though they initially examined the IGF-1R/IR expression in a larger panel of cells. Is it because these three cell lines express higher levels of IGF-1R and IR? If so, are they more or less sensitive to BMS-754807? Without that information, it is hard to assess whether the effects are specific to these cells.
4. The dose response curves in Fig 2A, 2B and 2C indicate that these cell lines are not particularly sensitive to BMS-754807, as the IC50s are 5 uM or above. Even at the highest concentrations they used, 10 uM, a significant level of cell viability remained. Thus, these results suggest that BMS-754807 is not particularly potent toward these cells, and such results seem to argue against the choice of IGF-1R/IR as targets in these cancer cells or this cancer type.
5. The paper has many minor editorial errors.
For example,
Line 15, “one the deadliest” is missing an “of”.
Fig 1A has a (B) embedded in the graph;
Line 266, BMS-75480 should be BMS-754807;
Line 268, “A scratch would closure assay…” should be “A scratch wound closure assay…”
Line 361, “Figure 6. Great anti-proliferative” should be modified by removing “great”.
Comments on the Quality of English Language
Good in general.
Reviewer 2 Report
Comments and Suggestions for Authors
The manuscript Inhibition of insulin-like growth factor signaling by the small- 2 molecule inhibitor BMS-754807 leads to improved survival in 3experimental esophageal adenocarcinoma is an important approach to understand the mechanism-specific efficacy of dual 468 IGF-1R/IR signaling inhibition with BMS-754807 as a mono and combined therapy with the chemotherapy drug nab-paclitaxel in experimental EAC. BMS-754807 as a monotherapy produced a clear greater in-vitro antiproliferative, pro-apoptotic, anti-motility effects and in-vivo antitumor efficacy by inducing enhanced in-vivo apoptosis with survival benefit compared to the observed in the control and the addition of nab-paclitaxel with BMS-754807 increasing these effects making them higher than the noticed after single agent therapy. The analysis or results support the importance of BMS-754807 alone and together with nab-paclitaxel; this treatment strategy needs to be tested in a phase 1 trial, EAC is a very serious and aggressive type of cancer and devastating for the patients.
Reviewer 3 Report
Comments and Suggestions for Authors
1. Author should elaborate the content for the molecule BMS-754807 and associated anticancer activities with supporting references.
2. The summary of work in the introduction section should be well descriptive and informative.
3. The experimental plan is not well structured. In vitro experiments are performed initially to screen how the cells respond to a tested compound. The inhibitory effects and concentration of a tested compound is evaluated and followed by molecular study further in vivo studies are performed to advance drug development studies. In the present study, western blot analysis has been performed initially and that is undecipherable. The treatment of cell lines was performed using 5 μM of nab-paclitaxel (NPT) and 10 μM of BMS-75807 (BMS). Authors should explain the basis for using these concentrations.
4. Cell viability should be included before molecular study. The concept of the experimental design is poorly understood. The growth medium should be mentioned instead of writing “proper growth medium”. How the concentration of nab- 139 paclitaxel and 1 μM of BMS-754807 for detecting the combination effects was selected and why assay was performed for 96 hours.
5. For in vivo study, authors need to mention whether they have performed acute or sub-acute toxicity for the selection of dose for treatment. If not, then what was the basis for the selection of doses for the treatment. What was the concentration used for combined treatment of nab-paclitaxel and BMS-754807.
6. Wound healing assay should be included under in vitro assays and not after in vivo study. The formula as mentioned for Wound healing rate should be elaborated. What was the basis for selection of 2 time points, 0 and 72 hours for screening the cell migration.
7. In Figure 2 and 5 check for the picture quality of graphs attached and uniformity of font size should be maintained for the figures in the entire manuscript.
8. In Figure 3, the band for cleaved-caspase 3 for the performed western blot analysis is not clear.
9. The figure legend of Figure 6 should be revised.
10. The conclusion part should be descriptive and included with the results obtained in the present study.
11. What is the basis for selection of different concentrations used for the treatment of EAC in the experimental section of Western Blot Analysis.
12. Authors have not taken any vehicle control in cell viability assay, why? What is the basis for selection of time point for treatment? This section should be revised.
13. What is the basis of selection for time points in scratch wound healing assessment? It is suggested to repeat the experiment and take pictures of different time points to validate the study.
14. There is a variation in magnification for the images recorded for scratch wound healing assay for 0 hours and 72 hours.
15. Authors cannot measure the rate of wound closure using formula mentioned in the material and method section. The result seems to be miss-interpreted, correct the result section for the same.
16. The conclusion section is poorly written, rewrite it in a meaningful way.
17. The experimental plan is not well structured. Without performing cell viability assay how authors have calculated the concentration of drug for the subsequent experiment ?
Comments on the Quality of English Language
Moderate editing of English language required
Reviewer 4 Report
Comments and Suggestions for Authors
The manuscript by Hassan and colleagues explores the small molecule inhibitor BMS-754807 on IGF signaling axis in experimental esophageal adenocarcinoma in vivo model. This small molecule inhibitor has been previously explored for its anti-cancer activities in several clinical trials but its role in EAC never explored. Moreover, its efficacy drastically improves with combination of NPT which makes it a novel for this study. In general, the manuscript is well written except for few inconsistencies. I have the following criticism for this manuscript.
1. Why only female athymic nude mice were used in the investigation?
2. Line 147 what is meant by quantifiable tumors? Mention the parameters used to define it.
3. Indicate the antibody dilutions used in this study for better reproducibility.
4. How the BMS and NPT dosage was fixed? And for combination what dosage was used, is this same as single dosage or any variations? Mention these in the methods section.
5. Be consistent in using NPT throughout the manuscript and no need to expand it in all the sections.
6. All the Westerns need to be quantified and should be statistically presented in the figures along with its molecular weight.
7. In figure 1 why there is difference in migration of phospho IGF and Total IGF between the samples?
8. Mention cleaved caspase 3 in the figure instead of “c-caspase 3”. Is the antibody being specific for cleaved caspase 3 or for total caspase 3 and presented only the cleaved/ degraded products?
9. Whether the cleaved PARP is for Asp 214? If so, mention this in the figure.
10. In figure 3 total IGF and total AKT blots are missing. Include this in the revised manuscript.
11. Mention the dosages used in the figure itself.
12. Figure 3 blots are dragged compared to figure 1. Be consistent in presenting.
13. Statistical analysis missing in figure 5 A and E.
14. In figure 5 B, C and D what does the asterisk indicates for the last bar (NPT + BMS) since this has been presented as two above and one below and no details presented in the legend.
15. How the proliferative index was calculated in figure 6B. Mention this in the methods or legend in detail.
16. In figure 6D error bar missing for NPT + BMS group.
17. Indicate molecular weights for all the Westerns presented in the raw images.
Reviewer 5 Report
Comments and Suggestions for Authors
Dear auhotrs,
you report that BMS-754807 when combined with 37 nab-paclitaxel enhanced those effects on inhibition of cell proliferation, on increment in cell apop- 38 tosis and on inhibition of wound healing BMS-754807 when combined with 37 nab-paclitaxel enhanced those effects on inhibition of cell proliferation, on increment in cell apop- 38 tosis and on inhibition of wound healing.
it is an important result, but I would like to known the impact of this data on the public health
Thank you
Comments on the Quality of English Language
Dear auhotrs,
you report that BMS-754807 when combined with 37 nab-paclitaxel enhanced those effects on inhibition of cell proliferation, on increment in cell apop- 38 tosis and on inhibition of wound healing BMS-754807 when combined with 37 nab-paclitaxel enhanced those effects on inhibition of cell proliferation, on increment in cell apop- 38 tosis and on inhibition of wound healing.
it is an important result, but I would like to known the impact of this data on the public health
Thank you